



# Defining and Analyzing the Frequency and Severity of Flood Events to Improve Risk Management from a Reinsurance Standpoint

*Elliott P. Morrill[12], Joseph F. Becker[2]*

[1]*Rosensteil School of Marine and Atmospheric Science at the University of Miami*
[2]*Guy Carpenter and Co. LLC*

*Correspondence emails:*

*Elliott.Morrill@guycarp.com*
*Joseph.F.Becker@guycarp.com*



**1.0 Abstract.**
The National Flood Insurance Program debt has accelerated research into private flood
insurance options. Offering this coverage begins with the ability to transfer the risk to the
reinsurance market. Within the industry perils such as hurricanes and earthquakes have standard
definitions but no such definition exists for flood. An event definition must examine the spatial
and temporal aspects of the flood as well as the complexities of individual events. In this paper
we were able to apply a data driven methodology to capture and aggregate flood peaks into
independent events. Analyzing both the HUC8 and HUC6 a total of 8,021 HUC8 events and
8,478 HUC6 events were recorded during the 15 water years used in our study. Each event was
characterized by duration, magnitude and severity. Focusing on the HUC8, events were unevenly
distributed nationally while severity was relatively evenly distributed. The goal for our study was
to take a method and be able to apply it to basins of varying characteristics. This framework
relied on the ability to analyze the individual processes related to each individual basin.



## 2.0 Introduction:

Throughout the world, flood events are one of the most destructive natural disasters. Floods occur for a variety of reasons, and risk factors such as total rainfall, soil types and land use can contribute to the complexity of events, in particular impacted area and event duration (Uhlemann 2010). Every year, major and minor floods contribute to economic and insured losses (Joyce 2014, FEMA). In the United States, the National Flood Insurance Program (NFIP) is the primary provider for residential flood insurance. Since its inception in 1968, the NFIP premiums have largely covered the amount paid out in losses (NFIP Act of 1968). However, the 2005 Hurricane season, including Hurricane Katrina, which was the costliest storm in the program's history costing more than 16 Billion USD, pushed the NFIP into debt (Fig.C1). The NFIP debt was exacerbated by the significant property damage experienced during Superstorm Sandy in 2012. Currently, the NFIP debt is estimated at $24 Billion (Joyce 2014).

This extreme debt has accelerated research into a number of different private flood insurance options. One necessary issue to address before primary flood insurance can become a more standard offering is the ability to transfer risk to the reinsurance community. A challenge specific to flood is the complexity of individual events. Unlike the perils with an unambiguous event definition, such as hurricanes and earthquakes, there is no standard definition for a flood event, which can range in length from hours to months. The problem for flooding is not specific to the United States. In fact, reinsurers have offered flood risk transfer products in Europe and Asia for a number of years. For example, (re)insurers in Spain have provided flood insurance since 1971 (Barredo et al. 2012). Typically, reinsurance contracts define a flood event using an hour's clause ranging between 168 hours in the UK to 504 hours in Germany. Using the hour's clause insurance companies are able to aggregate claims during this period of time to limit



cumulative losses from multiple events (Munich Re. 2005). Defining events this way allows for
providers to aggregate claims that can be associated with the same temporal event.

50        However, the hour's clause definition lacks the ability to discern between the shorter and

longer events. Not all events can fit into a single defined time frame. If there are multiple short
duration events occurring in quick succession then the claims from those events maybe
aggregated together. The hour's clause also lacks the ability to determine spatial aspects of each
flood event. If events occur within the same window of time but in two different areas those
flood are still attributed to one event. Aggregating these events limits the ability to understand
the spatial extent based on impacted areas and the severity of each of the individual flood
occurrences.

58        While research into flood event definitions is accelerating, it is not a novel topic.

Research into event definitions has primarily focused on single site analysis (Bačová-Mitková &
Onderka 2010, Mallakpour & Villarini 2016 and Kahana et al. 2002). However, as flood events
are spatially complex, they often impact many locations limiting the use of single site definitions
for reinsurance contact definitions. When events impact larger areas, multiple locations or entire
basins, there is no method that can properly group flood peaks to the same event.

64        Public entities have complied databases of flood occurrences to assist in frequency and

severity analyses (NCDC). One goal of this type of analysis is to determine if floods are
occurring more often and with increased severity due to climate change or other anthropogenic
causes (Himmelsbach et al. 2015).  Public databases are comprised of documentary sources and
trained spotter observations (NCDC, EM-Dat, and DFO). The major downside of using this type
of database to assist with reinsurance contracts is that they are based on subjective measures such
as spotter definitions. Definitions follow a series of guidelines but varying flood characteristics



between regions can categorize flooding differently between these two regions. Variations in
categorization have an impact on event durations and impacted areas. In addition to the
definitions themselves, trained spotters respond to citizens reports of the peril. Depending on the
area, what is considered abnormal flooding, in terms of standing water or bankfull discharge,
may be reported in one area compared to another. For example an area such as Florida
experiences significant precipitation year round which may contribute to minor flooding that is
considered normal and thus not reported. However in an area like Los Angeles that similar minor
flooding may be reported, which affects the frequencies of flooding in each area. Another source
of flood occurrence information is using a documentary source, which involves examining media
sources as well as government reports to comprise a set of occurrences across a state, country or
globe (Himmelsbach et al. 2015 and Doocy et al. 2013). These sources rely heavily on the
quality of the reporting, using the reports to assign severity and frequency estimates to cover an
expansive region.

Relying on the quality of the reporting can lead to inconsistencies in what is reported and

how it is reported. In a number of areas you can have two sources that can report statistics about
an event that are drastically different. From those reports determining which is the most accurate
becomes a challenge. Another issue with secondary sources is being able to define event
duration. In many cases the reports cover the first instance of flood and damages associated but
do not report the flooding on subsequent days defining the event duration. Spatially defining the
event presents challenges. Not all events are reported equally across all areas. Secondary sources
will primarily focus on the most severely impacted areas, but that provides a small picture of the
entire event.

EM-DAT and National Climatic Data Center (NCDC) *Storm Data* databases are the two



that are most commonly used datasets for this type of analysis. EM-DAT uses official records of
areas affected, persons killed, disaster declarations issued and calls for international assistance
made (EM-Dat, Doocy et al. 2013). The NCDC *Storm Data* database is a compiled set of
observations from National Oceanic and Atmospheric Administration (NOAA) trained spotters.
NCDC events are categorized by county and then separated by dates (Dobour and Noel 2005,
Gaffin and Hotz 2000). EM-DAT catalogues events by year with summary statistics detailing
frequency and overall event impacts (i.e. deaths and losses) from that year. Such summary
statistics include injured, affected, total deaths and total damage. Both methods contain a number
of different biases preventing use in reinsurance contracts including population biases, frequency
biases and reporting biases. Due to the incomplete and often inconsistent reporting,
implementing this method to formulate an event definition for reinsurance contracts presents a
challenge. Despite their limitations, these datasets are useful first checks when developing a
more robust method to define flood events.

Many authors have shifted toward a data driven approach using the peaks over threshold

analysis to examine changes in flood event frequency (Mallakpour and Villarini 2016, Bačová-
Mitková & Onderka 2010), as well seasonality (Black and Werritty 1997). A data driven
approach allows for the definition of an event to encompass a variety of basin characteristics.
Authors choose a somewhat arbitrary threshold where if a peak observation exceeds the
threshold, it is considered to be a peak over threshold (POT). A subsequent step for this method
was to determine a metric for identifying independent peaks. Varying windows of time were
used to identify the independence between the individual POT. Mallakpour and Villarini used an
arbitrary window of 15-days, where any peak that occurs within this period is considered a single
event. Black and Werrity determined their window by calculating the "time to rise" and



identifying when the discharge dropped below 2/3rds of the previous peak. Authors using these
windows then looked at all individual peaks occurring within these windows to attribute them to
the same event.

Site specific event identification is the base in developing a consistent method of event

identification. However, our method will address the window of independence through an
observational approach. Event independence should not be based on a standard window
(Mallakpour and Villarini 2016). It must be based on how each site reacts to the flood waves.
Implementing a concept similar to time to rise and a drop in discharge (Black and Werritty 1997)
was the first of many steps taken toward resolving this. The window must be established to cover
the time before and after a peak, as previous peaks have an influence on subsequent peaks.
Incorporating this into our definition will reflect the individuality of each site and the flexibility
of our definition to cover a wider range of sites.

The primary goal of this research is to expand our definition to an entire basin or

catchment area. These regionally impacting events are titled basin or "trans-basin" events (Nied
et al. 2014, Uhlemann et al. 2010). Both papers used the POT method as well. Starting with a
single site, individual events were identified (Uhlemann et al. 2010) and then all mutually
dependent events were identified from a moving temporal window. The window defined from
previous literature provides a solid structure but categorizes catchments and basins into an all-
encompassing time frame. A more basin specific time frame is measurable and would not
underestimate the smaller basins or overestimate the larger basins.

This paper seeks to define events through a data driven approach aimed at accounting for

the individuality of flood waves and the basins they impact. Our main goal is to develop a
consistent definition in order to examine how frequency and severity vary regionally. Looking at



frequency regionally provided us with a clearer picture of the specific areas that were more at
risk for flooding. Severity allowed us to look at how areas with similar frequencies were
experiencing events in terms of impacted areas and overall magnitude. Severity will factor into
future implementation of risk mitigating factors that can look at two areas and determine the
steps needed to protect a certain area. It also allowed us to determine if our method is
representing more local or extreme flooding across the various basins.

Methods implementing the hour's clause or standard event windows lack the ability to

interpret how each individual flood wave progresses. Understanding the individuality of the
flood is the basis for how our method will tackle a standard event definition. This paper will be
structured as follows: Section 2 will cover the data availability as well as the data selection
process along with which tools were used to analyze the data. The concepts that feed into our
method as well as our method itself will be discussed in Section 3. Section 4 will provide the
results of the analysis from our methodology with comparisons to methodologies exhibited in
previous research. Section 5 will provide the discussion and concluding remarks regarding our
results within this study.
**2.0 Site Selection:**

This research focuses on expanding the definition of a flood event from an individual site

to river basin. As this research focuses on the United States, USGS daily flow gauges stations
were used to identify individual sites and USGS Hydrological Unit Codes (HUC) were used to
define river basins. River basins can be defined in a number of ways and determining the
appropriate size can be a non-trivial task. For use in reinsurance contracts, river basin should be
defined in such a way that flooding events within a portion of the basin show a correlation to
events in other portions.  A river basin needs to be defined in such a way that we can see how



flood waves impact the basin and not individual sections of that basin. The USGS HUC codes
follow the Pfafstetter Coding System meaning that each unit code is delineated in a hierarchical
fashion. Drainage areas are defined on a continental scale and then divided and subdivided into 6
levels. Each level is associated with number of digits corresponding to size. Digits range from 2
– 12, largest to smallest (USGS), with the 8/6 digit HUC's being used. These two levels were
chosen as they were felt to best represent how flood waves would impact a basin. Daily mean
discharge as well as Annual peak streamflow was used for all sites, which provided data for
those parameters.
From all available HUC's, sites and basins were selected based on a number of selection
criteria. The first criteria removed sites with less than 5 years of daily discharge data. The second
criteria required sites to occur along natural rivers and streams; gauges impacted by reservoirs
and other impediments to natural flow were excluded. Following site removal, HUC's with less
than 5 sites were excluded. Finally, HUC's were required to have at least 3 sites that overlapped
with 70% of the data during each individual year that was examined. Due to the nature of our
method seeking to aggregate peaks from multiple sites, the sites needed to overlap or else that
method would be looking primarily at individual site events instead of the basin events. Of the
2,300 HUC8's and 387 HUC6's available, 466 HUC8's and 276 HUC6's were used (Fig.1) with
a total of 3,164 and 4,920 gauge stations within the HUC8 and HUC6 respectively. Both HUC
sizes were analyzed for initial frequencies and the most applicable HUC was chosen for
subsequent analyses.
**3.0 Methodology:**
Daily discharge data from 8,084 river gauge stations was obtained from the USGS
(*http://nwis.waterdata.usgs.gov/nwis/dv/?referred_module=sw*). A study period of 15 water



years between 2000 and 2015 was selected for this analysis. Initial attempts to expand the period
of analysis severely reduced the number of basins that met the criteria for analysis. The peak
over threshold method outlined in Uhlemann et al. (2010) was conducted on all basins that fit the
criteria for analysis. The peak over threshold method consists of identifying individual
observations over a specified threshold within a particular time window. The procedure was split
into 4 major steps: (1) identifying peaks occurring at each site within each basin and the
subsequent peaks over threshold; (2) applying a window of independence at each site to
determine independent site specific events; (3) compiling all independent site specific events and
applying a secondary window of independence to determine independent basin specific events;
(4) applying multiple characteristics to determine a severity score to compare differing events
from one another.

The first step involved selecting a minimum threshold. The median of annual maximums

was chosen as the threshold in which a flood peak must exceed. The median of annual
maximums was chosen because it corresponds to the 2-year quantile, or Q2. Uhlemann et al.
(2010) states that the "Q2 is a rough estimation for bankfull discharge on naturally occurring
streams." For sites with at least 5 years of annual peak streamflow data, their Q2 was calculated
by taking the median across the entire time series. Sites with less than 5 years of data had their
respective Q2 calculated from the annual maxima obtained through their daily discharge time
series. As peak discharges are determined by instantaneous measures, small catchments can
exhibit extreme values, which are rarely observed in the daily record. The extreme values may
lead to a minimum threshold that may not be a representative measurement of flooding for that
catchment area. The distinction between the use of the annual peak streamflow and daily mean
discharge data was made to ensure that the threshold was not impacted by drastic variations





within the annual maximum during a short period of time. The discharge at each of the peaks
recorded, were then compared to their respective sites Q2 value to determine all of the peaks
over threshold.

The next step in identifying site specific events is to determine a time criteria that defines

independent site events. Two metrics were calculated for all peaks over threshold to determine
the duration of each event: base to peak (BtoP) and peak to base (PtoB). Base to peak is the time
it takes for the discharge to reach the peak after it has crossed the minimum threshold. Peak to
base is the amount of time it takes for the discharge to return to the minimum threshold
following a peak (Fig.2). In the case where there are multiple peaks before the discharge returns
to base, the peak was selected as the observation that experienced the maximum discharge. Each
peak over threshold has a unique BtoP and PtoB that could have a significant range. To
standardize the windows of independence for each site the median of both metrics was calculated
and then the peaks start and end times were recalculated. Our window of time was aimed at
eliminating the extreme events on either end of the temporal distribution to determine a window
that reflected the time it would take for a flood wave progress through a site.

After the windows were recalculated, combining peaks with overlapping or consecutive

windows into a single site specific peak consolidated peaks. All peaks over thresholds with
windows that did not overlap were treated as independent events. Each event was characterized
by, site number, start time, peak time, end time and peak discharge. For the peaks, which
overlapped, the start time was defined as the earliest start day and end time was the latest end
date. The peak discharge from each event was then scaled by the Q2 at each site. Scaling each
peak discharge reduced the impact of catchment size when comparing magnitude of discharge
and made the different sites comparable.





A similar methodology of consolidating overlapping observations was applied to define
basin specific events from the site specific events (Fig.3). The basin specific events used the start
and end time of each site specific events that occurred within the basin. If the windows of time
between the start and end of the site specific events overlapped or were consecutive (i.e.
occurred within 1 day of another peak), then these events comprised one basin specific event.
The start of the event was the earliest start time recorded at any site and the end of the event was
the final end time recorded. Each event was defined by start time, end time, peak time, and peak
discharge for all events from the desired HUC's.
The final step involved determining a severity score for each basin event. Defining
severity allowed us to compare areas of like frequency. From these we were able to see the
certain areas that are more vulnerable during flooding. Severity scores in future analyses will
also factor into pricing of reinsurance contracts. Severity of each event was designed to include
elements of the spatial extent as well as the magnitude of the flooding experienced in the basin
by the affected sites during each event. The severity score represents a number between 0 and
infinity where the high value indicates a more severe event. The impacted area was defined as
the number of sites within the desired HUC, which recorded a peak over threshold during the
event. Total discharge was the sum of the discharges, scaled by their corresponding minimum
threshold, observed at all the impacted sites. Severity was calculated by taking the sum of all
scaled discharges and dividing by the total number of sites within the basin, *EQ.A 1*. If a site was
impacted more than once during a basin event, the maximum-scaled discharge was selected to
calculate the severity score. Scores less than one are expected when looking at the minimum
threshold as it represents small scale and localized flooding, in terms of discharge and the
percentage of sites it may impact within the individual HUC.



From the analysis, we compared the HUC6 and the HUC8 to determine which size basin
was more appropriate for our method. For each HUC aggregation, frequency, event duration and
severity distributions were examined. Two comparisons were made to the NCDC *Storm Data.*
The first method looks at all reports of flooding and aggregates them by county. The second
method used a standard 13-day independence window, 3 days pre-peak and 10 days post-peak
(Uhlemann et al. 2010). A standard window was used because the NCDC observations are
unable to provide a site specific window of independence.
**4.0 Results:**
A total of 8,021 and 8,478 events were calculated for basins defined by the HUC8 and
HUC6 respectively. *Table.B1* provides the frequency summary statistics for both the HUC8 and
HUC6 basins. Comparing the frequency distribution of events between the two selected basins
sizes suggests that frequencies within basins defined by the HUC6 are higher than frequencies
defined by the HUC8 (Fig.4 & Fig.5). This comparison is important because the aim of this
paper is to define events at a basin level suitable to use in reinsurance contracts.
To test for this, we examined the impact of number of sites within a basin on the number
of events for basins defined by the HUC8 (Fig.6 left plot) and the HUC6 (Fig.7 left plot). In the
basins defined by the HUC8, there is a gradual increase in event frequency as the number of sites
increases, however, there is a more dramatic rise when the basin is defined by the HUC6
indicating that there is stronger positive correleation between the number of sites and event
frequencies. For each HUC, there was no interaction between the size of the catchment and the
number of events (Fig.6 andFig.7– right panels).
Nationally, the median frequency of events HUC8 basins was 1.00 events per year while
the mean was 1.15 events per year (Fig.8). This frequency varied regionally with some areas





experiencing higher frequencies (Fig.9). Notable population centers that experience elevated
frequencies include the Upper Midwest (south of Lake Michigan), Southern California and
Southern Florida. While these population centers experienced elevated frequencies, there does
not appear to be a population bias throughout the study. For the HUC6 basins, the median
frequency of events was 1.87 events per year with a mean of 2.05 events per year (*Fig.10*).
Similarly to the HUC8 basins, the frequencies varied regionally with some areas of elevated
frequencies (*Fig.11*).

To investigate how event duration varies nationally, we calculate the mean event duration

for each basin. Nationally, the mean event duration ranged from two to 79 days for the basins
defined by the HUC8 and two to 73 days for the basins defined by the HUC6. The mean event
duration for 95% of HUC8 and HUC6 basins is less than 14 and 17 days respectively (Fig.12 and
Fig.13). The minimum event duration was two days and was observed at 336 HUC8's and 227
HUC6's. The maximum event duration for HUC8's was 232 days and occurred in the 10160003
basin. For HUC6 basins that maximum event duration was 237 days occurring in the 101600
basin.

Figure 14 represents two sites that reflect longer recession periods following their peaks.

With a data driven approach identifying the generation and recession of the events, certain
extreme events may show extreme durations based on their observations. The extreme durations
are a reflection of the minimum threshold as well as the hydrological processes at hand. Looking
at the two sites, the left is located in South Dakota and the right is located in Florida; both of the
extreme events that are observed have certain factors that impacted their recessions. The site in
South Dakota experienced an event that was impacted by the melting of an ice jam represented
by the quick generation. Following the melt there was a significant rain event as well as a release



of water from a dam further upstream. The site on the right is located on a natural tourist spring.
These springs contain a significant amount of ground water. Following an intense rain event the
buildup of water caused the increased recession. The duration of the events represent the
observations at each site so based on our definition we can see long event durations. These long
durations are slightly longer than we would expect and further analysis will be conducted to
examine changes to the minimum threshold. While a majority of the durations reflect reasonable
time frames for flooding events that exceed the Q2 it is important to note that the method might
not be appropriate for all streams.

When looking at the distribution of severity scores there is a slight skew towards the

extreme events. Severity scores ranged from the least severe, 0.032 to the most severe, 26.9
(Fig.15) with a median severity score of 0.32 and a mean of 0.57. While the range in severity
scores is quite large, a majority of the events received a score less than 1. Regionally the severity
scores are generally distributed evenly throughout the country (Fig.16). There appear to be
pockets of higher severities but across the country there does not appear to be a pattern within
the regional distribution.  While it is evenly distributed regionally, within the regions we can see
the wide range in severity that was observed in the distribution of frequency.

Finally, comparisons were made to other methodologies applied to the same dataset as

well as other publically accessible datasets. The first comparison examined a method used by
FEMA to estimate floods using NCDC Storm Events Database (Fig.17). The distribution of
events was broken down into total event frequency by county ranging from one to 4,114. While
the trained spotters follow guidelines in identifying events, the method lacks a way to group
events. The inability to group events that would otherwise be considered a single event, leads to
an overestimation of events. This overestimation is evident when it is noted that the maximum



frequency of events for a specific county was 4,114.

The final comparison was made to the NCDC applying a 13-day standard window. While

the NCDC map provides a more complete national coverage two patterns occur (Fig18). Within
the 5-boxed areas, either the NCDC frequency is far greater or the daily discharge frequency was
far greater. For example, in Florida, we see frequency range from 6 to 25 events for NCDC
observations but events observed through daily discharge range from 26 to 45. The opposite
occurs in Missouri with NCDC estimates ranging from 16 to 85 events with events observed
through daily discharge ranging from 6 to 15.

From these estimates there is no obvious reason for the discrepancies in frequencies but

we can speculate. For example Florida experiences significantly fewer events using NCDC data
than the daily discharge data. A possible explanation could be how trained spotters define events.
An area in Florida may experience a peak over the threshold triggering our event definition, yet
that peak may not be recorded as an NCDC observation based on the spotters perspective.
Another reason could be due to the fact that these trained spotters respond to citizen's reports
and, due to the frequency of flooding in an area like Florida, the citizen may not call and the
peak may not be recorded.

However a similar thought process can be applied to our threshold selection. As stated

the minimum threshold was selected as a representation of bankfull discharge. While this
assumption was the basis for our method, in certain areas it is conceivable that the threshold may
be lower than bankfull discharge which could possibly lead to an over estimation of flooding
events in certain areas. There is no certain explanation for the discrepancies in the results. With
no certain explanation for the results from this comparison, the assumptions that define the
compared methodologies will be explored in future analyses.





## 5.0 Discussion and Conclusions:

This study was able to provide a data driven approach in attempts to solve the issues of inconsistent event definitions within the (re)insurance industry. We derived a methodology based on a peak over threshold analysis that was able to capture and aggregate multiple occurrences of flooding at various locations. Using physical assumptions, our minimum threshold and window of independence were able to capture each individual sites reaction to passing flood waves. An approach identifying windows based on the impacted site allows for each site to represent their individual characteristics of flooding rather than applying standard metrics throughout. Each event was defined through their duration, impacted area and magnitude. The development of a severity index examines overall impacted areas as well as individual flood magnitudes.

Analyses were conducted on both HUC8 and HUC6 to determine which size of Hydrological Unit Code was more applicable for further analysis. 8,021 HUC8 and 8,478 HUC6 events were identified during our study. Understanding the applicability of different basin sizes is important because it aids in our main goal of applying a consistent definition to reinsurance contracts. From our definition our goal was to understand the frequency that represents an entire basin or area. We also hope to use the definition to define a parametric trigger or an alternative form of defining the event. All of this is possible when we know what basin size is the most applicable. The HUC8 was chosen as a more applicable basin size as it was a better representation of site interaction during flooding events.

Nationally, there are areas with large discrepancies between the HUC6 and HUC 8 frequencies. One explaination of this discrepency is represented by HUC6 (071200) Fig5. The area of this HUC6 is 28,309.78km$^2$ and contains 6 HUC8s. The annual frequency of events of the HUC8 ranges between 1 and 2.33, while the HUC6 produces 5 events per year. Although it is



expected that the larger basin will have a slightly higher frequency due to some events occuring
in one part of the basin and not impacting the other, a more than doubling of events per year
indicates that a large number of events do not interact with other sites in the basin. This lack of
interaction is inconsistent with the goal of this research to identify basinwide event frequencies.
The inconsistencies and lack of interaction are represented by the relationship between site count
on frequency (Fig.6 & Fig.7).

We found that HUC8 frequencies are relatively normally distributed but are unevenly

distributed regionally. For all HUC8's a median of 15 events (1 event per year) and mean of
17.21 events (1.14 events per year) were recorded. In a number of areas there were pockets of
elevated frequencies. Durations for all events ranged from 2 – 232 days with a mean duration of
6.34 days. The wide range of event durations prompts further investigation into events with
durations in the positive tail of the distribution. For example, we considered two HUC8's, one in
South Dakota (10160003) and another in Florida (03100207), that are impacted by natural events
leading to longer durations. Some sites within these two basins were affected by ice jams as well
as natural springs, which have contributed to significant recessions of their events. While these
events are natural, the resulting event durations should prompt examination into the selection of
thresholds for the sites, as an assumption of bankfull discharge might be slightly lower than a
threshold that produces flooding.

Severity scores calculated for all events in the dataset showed a slight skew toward the

more extreme events. The smaller and local events are represented by the median of 0.32 and
mean of 0.57, as we can expect events slightly above the threshold to not necessarily affect all
the sites in the basin, producing a score less than 1. Regionally severity is relatively evenly
distributed nationally.



With a data driven approach to our methodology, a focus on the individual site
parameters shifts the focus from generalities about events to site specific understanding leading
to an applicable method regionally. A fundamental aspect of this research is to understand spatial
extent of flooding and we were able to expand from single gauge stations to entire basins. The
data driven approach allowed us to apply the methodology to a number of basins with varying
characteristics. The final advantage to our method is that when looking at flood severity we do
not look at exclusively magnitude but the addition of spatial extent adds an element to
differences in severity regionally.
While there are a number of advantages that come from this method, relying on public
data have revealed drawbacks in its application. Being a data driven method limits our ability to
estimate frequencies in areas that do not have data. Across all USGS gauges there is no
uniformity in data availability for number of years or number of stations within a basin. Through
our site selection process we were only able to use 25% of all available HUC8's, which limits
national coverage in our estimates.
The minimum threshold for flooding is based on the assumption that it is a representation
of bankfull discharge; in certain areas this may not be accurate. Riverbanks are not uniform so
how bankfull discharge is recognized at each site is dependent on that location, which may lead
to underestimation or overestimation of flood stage at that site. The final drawback we observed
was that when taking the median of the BtoP and PtoB slight variations in the event window
occurred on the more extreme events. Instead of median other statistics will be tested to
determine the most applicable way to represent the basin flood generation and recession.
For further research a comparative analysis will be conducted altering the threshold to
examine how that might affect frequency as well as severity. Increasing the time frame will also



provide insight as to whether or not this 15-year period is representative of the entire time frame
of data or if we see a significant increase in events during certain subsections. Seasonality tests
will be run to observe areas more frequent and more severe times of year which may also
provide insight for risk managers. The final test that will need to be conducted is a sensitivity
analysis on the threshold selected to prove which threshold is the most reasonable for an analysis
such as this.



**Code Availability:**

All calculation and download scripts have been included in the supplemental folder. All scripts were written using R-Studio.

**Data Availability:**

All data is publically available from the NCDC Storm events database as well as the USGS stream gauge data sites. A list of sites and a list of the years used will be included as well as the compiled file of the data, added to the supplemental files.
ftp://ftp.ncdc.noaa.gov/pub/data/swdi/stormevents/csvfiles/
https://waterdata.usgs.gov/nwis/uv

**Appendices:**

**Team List:**

Elliott P. Morrill
Joseph F. Becker

**Author Contribution:**

E. Morrill and J. Becker designed the methodology. E. Morrill wrote and executed code to carry out the methodology. E. Morrill performed the manuscript with help from other authors.

**Competing Interests:**

The authors declare that they have no conflicts of interest.

**Disclaimer:**

The opinions expressed by authors contributing to this journal do not necessarily reflect the opinions of the Hydrology and Earth System Sciences Journal or the institutions with which the authors are affiliated. The data and code used within this research is a property of Guy Carpenter and Co. LLC.

**Acknowledgments:**

I would like to acknowledge the support of Guy Carpenter LLC and the Nat/Geo group within the Analytics Department. I would also like to thank my advisors from the University of Miami, Igor Kamenkovich, David Letson, Roni Avissar for supporting me during my time at The University of Miami as well as my time at Guy Carpenter and Company LLC.



**499**   **Appendices:**
**500**
**501**   **Appendix A.**
**502**

$$\text{Severity} = \dfrac{\sum Q(i)_{Scaled}}{\text{\# of Sites (HUC)}}$$

**503**
**504**                          *EQ.A 1. Severity Score*
**505**
**506**   **Appendix B.**

| HUC | Total HUCS | Selected HUCS | Minimum Freq. | 1st Quantile | Median Freq. | Mean Freq. | 3rd Quantile | Maximum Freq. |
|-----|-----------|---------------|---------------|--------------|--------------|------------|--------------|---------------|
| **08** | 2300 | 466 | 0 | 10 | 15 | 17.21 | 21 | 63 |
| **06** | 387 | 276 | 0 | 19.75 | 27 | 30.72 | 38 | 153 |

**507**                *Table.B1. HUC8 and HUC6 Frequency Summary Statistics*
**508**
**509**   **Appendix C**

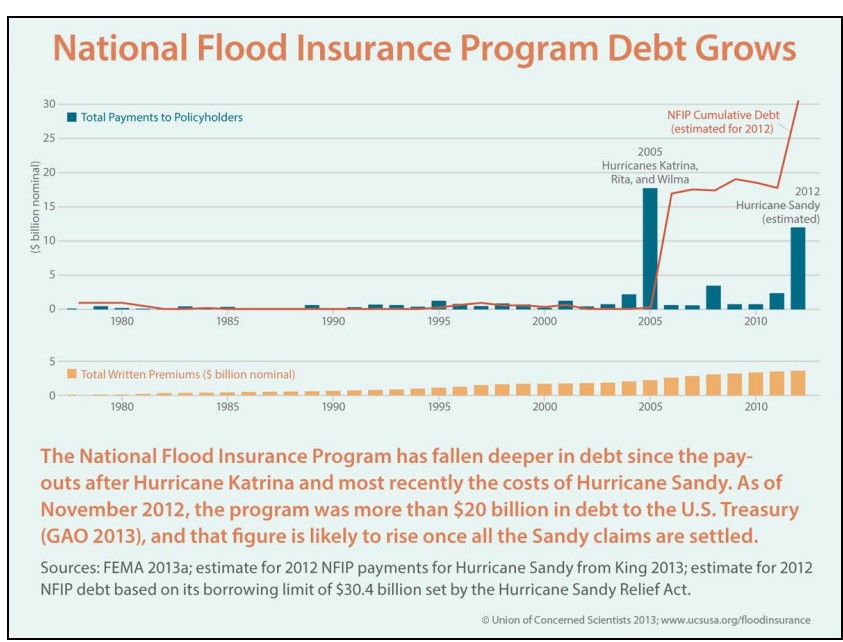

**510**
**511**        *Fig.C1 NFIP Cumulative Debt, Total Payments and Total Premiums, 1978-2012*
**512**
**513**
**514**
**515**
**516**
**517**
**518**
**519**
**520**
**521**



**Figures**

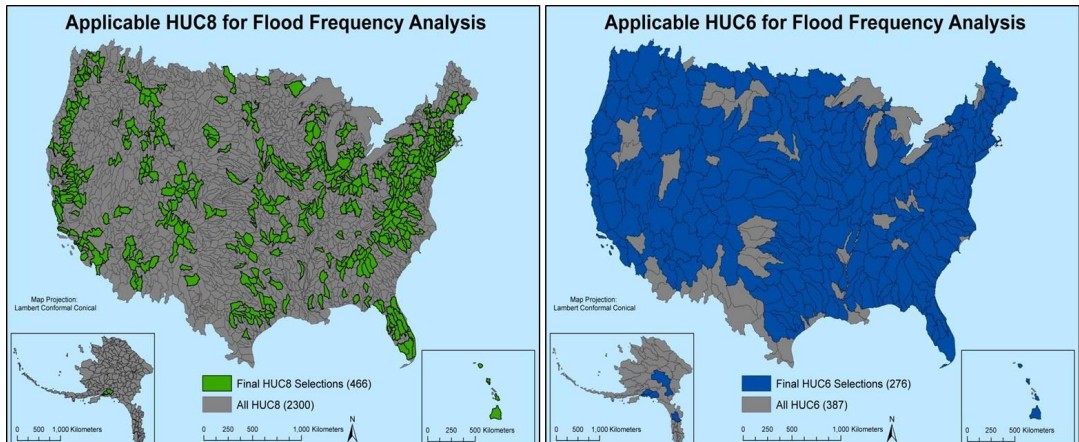

*Fig.1. A map of selected HUC8 and HUC6*

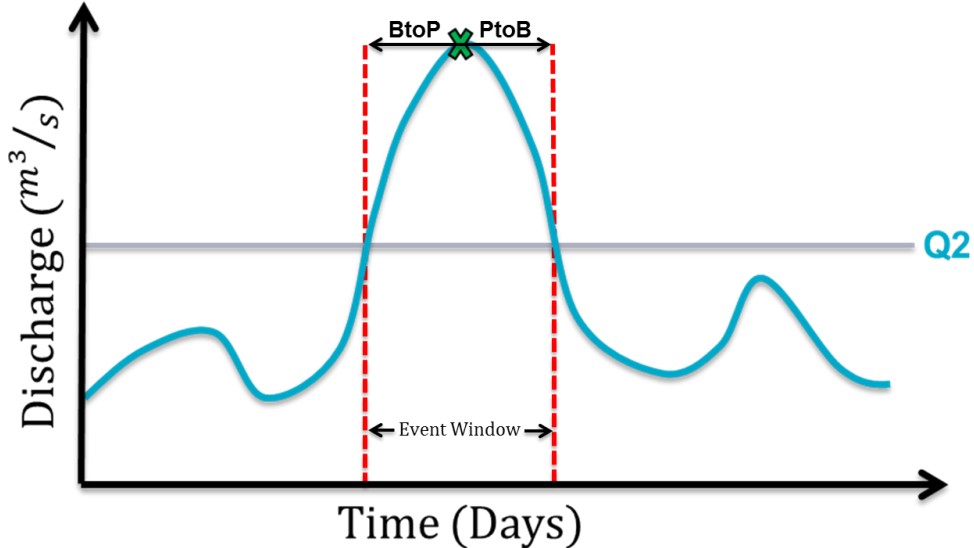

*Fig.2. Site Event Identification*




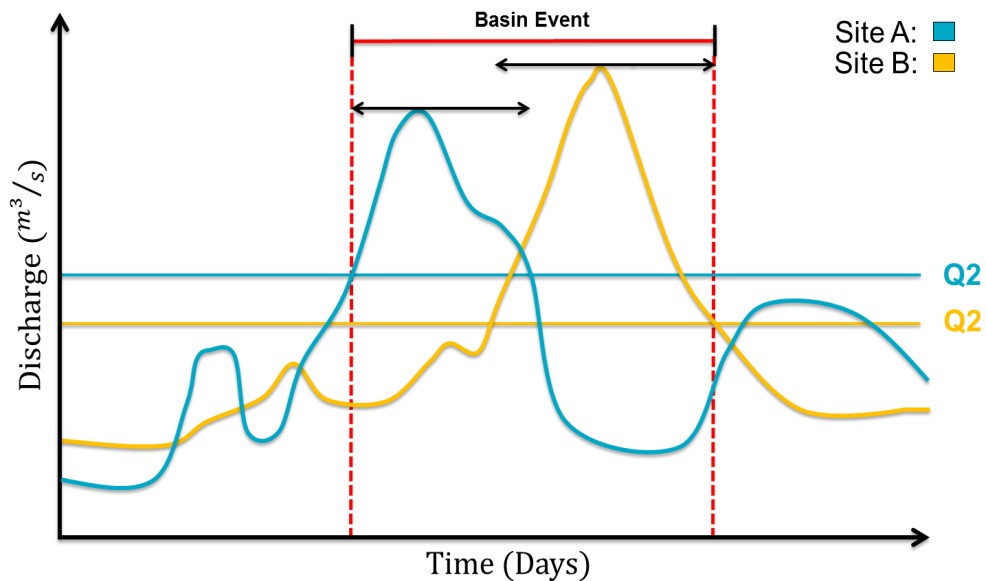

***Fig.3. Basin Event Identification***

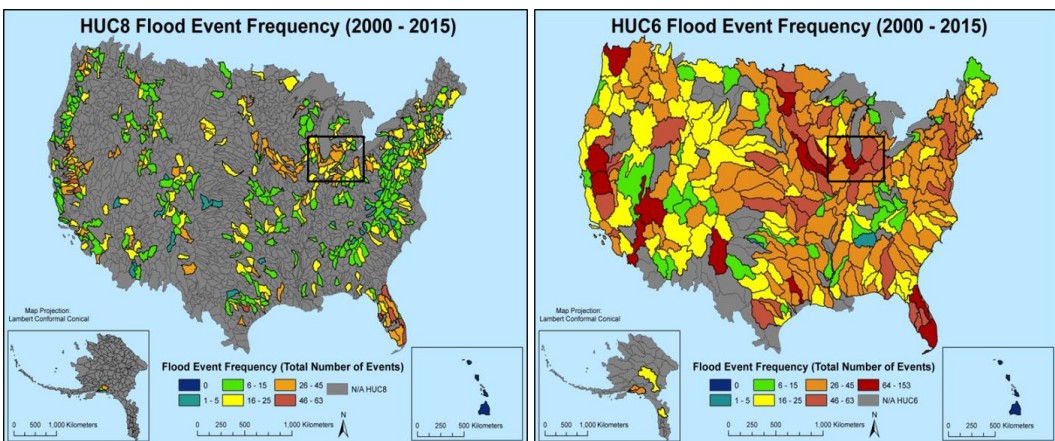

***Fig4. HUC 8 and HUC6 Frequency Comparison, National***



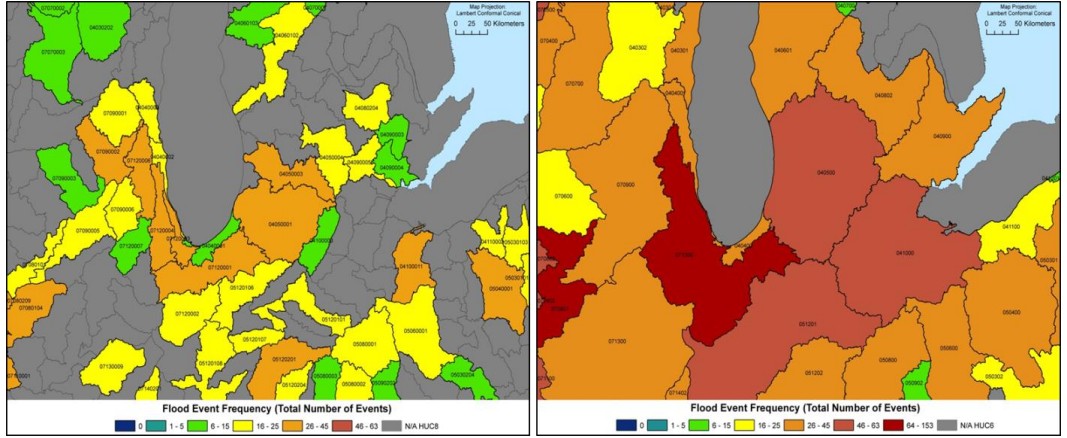


***Fig.5. HUC 8 and HUC 6 Frequency Comparison, Upper Midwest***

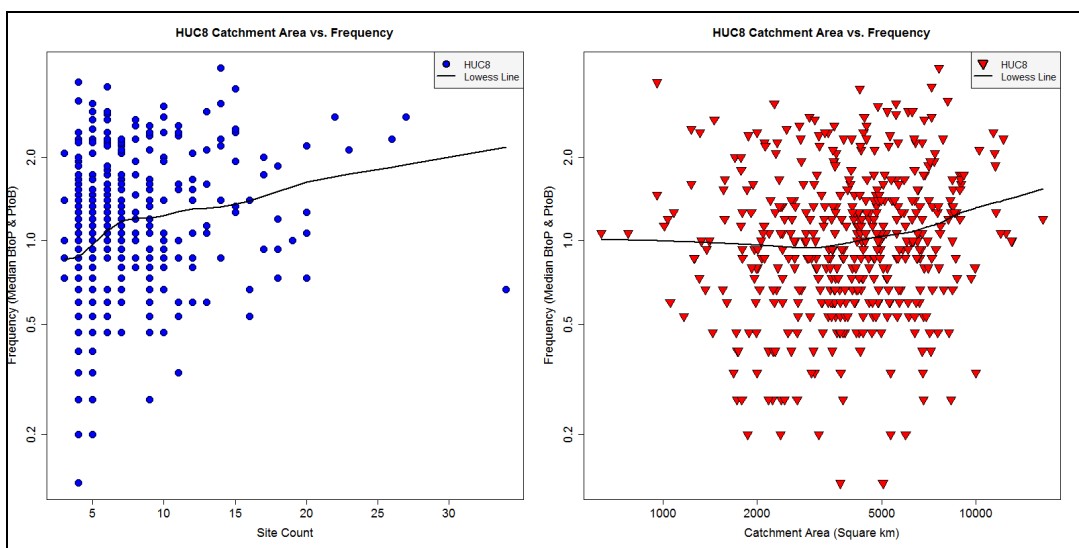


***Fig.6. HUC8 Site Count vs. Frequency & Catchment Area vs. Frequency (y-axes are in log scale)***




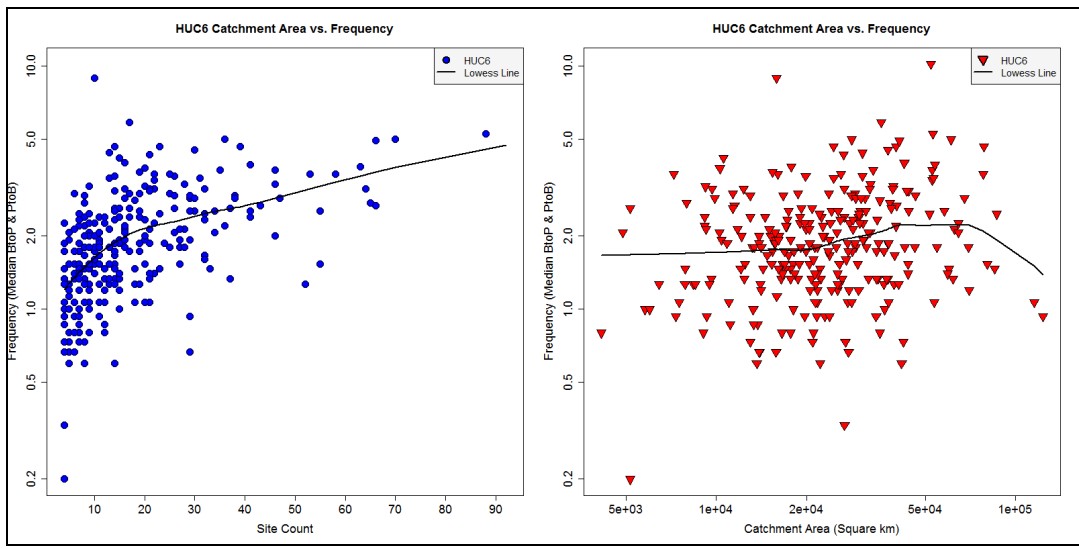

*Fig.7. HUC6 Site Count vs. Frequency & Catchment Area vs. Frequency (y-axes are in log scale)*

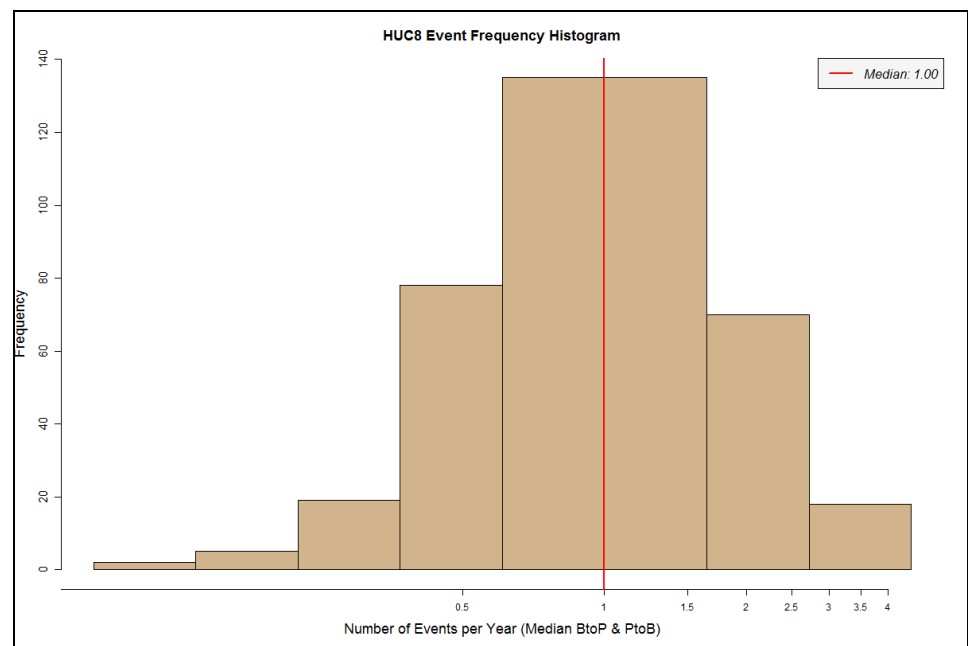

*Fig.8. HUC8 Frequency Distribution*

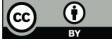



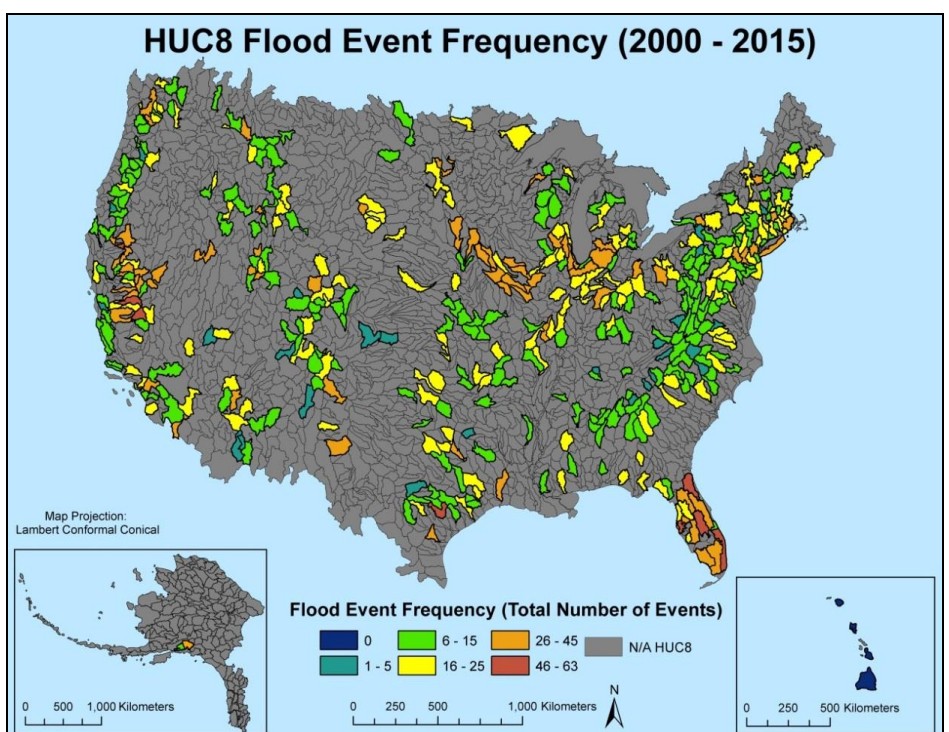

*Fig.9. HUC8 Regional Frequency Distribution*

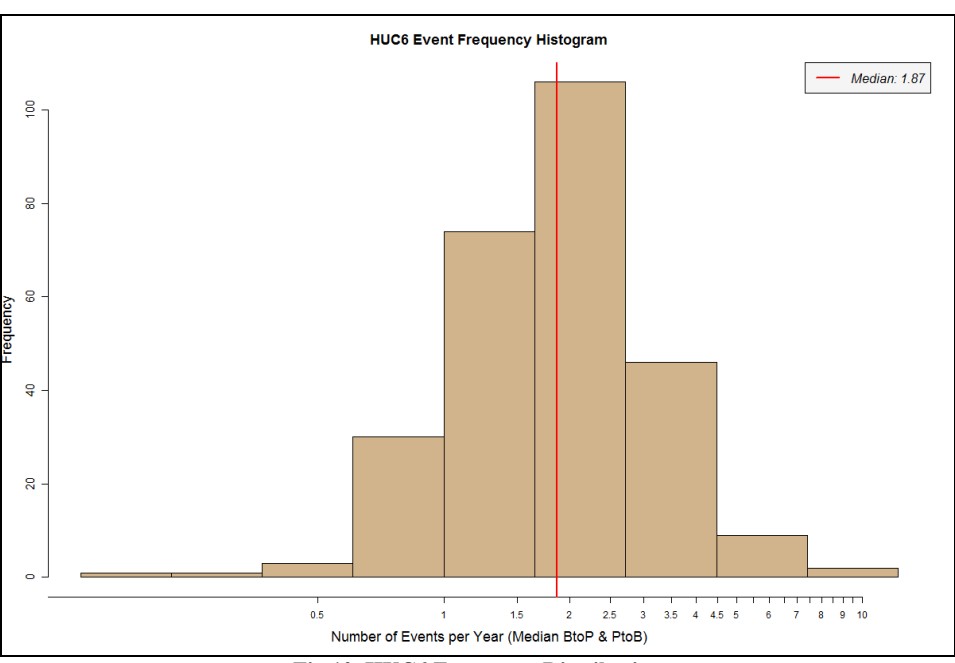

**Fig.10. HUC6 Frequency Distribution**



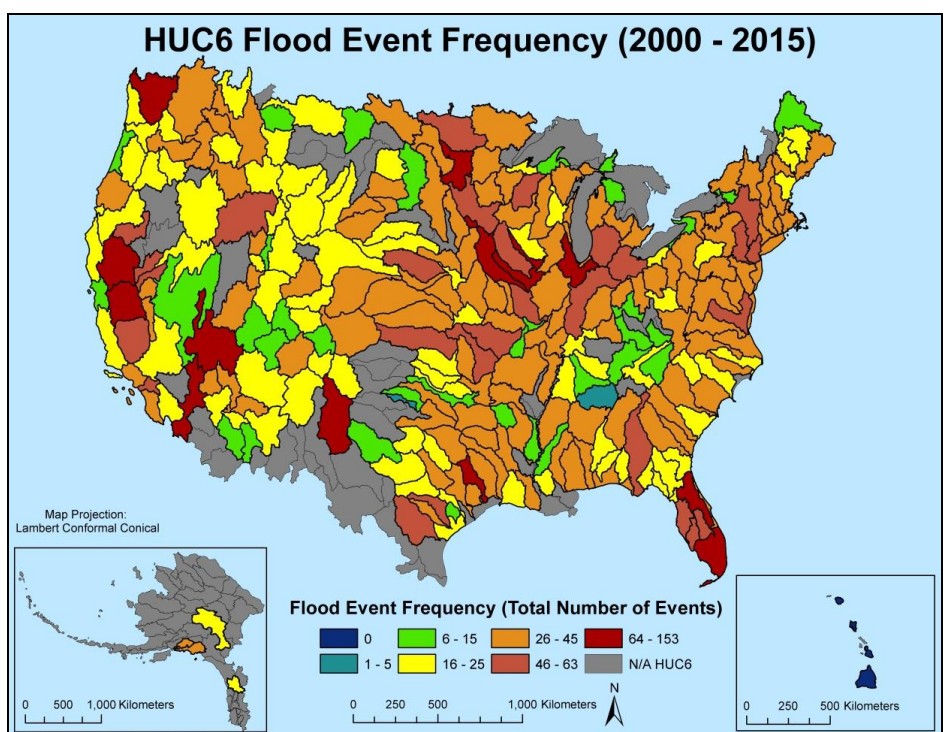


**Fig.11. HUC6 Regional Frequency Distribution**

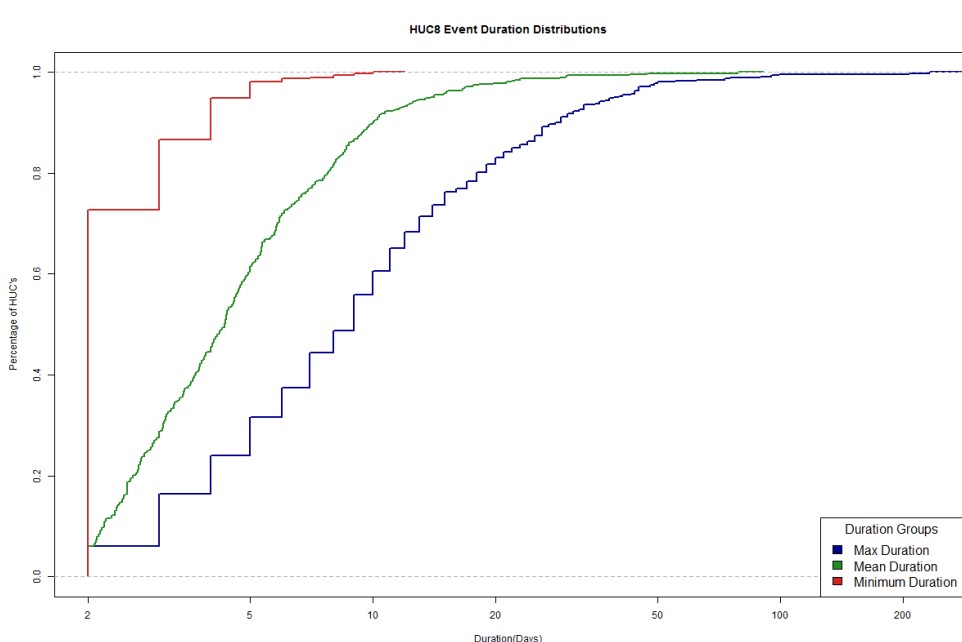


*Fig.12. HUC8 Event Duration CDF*




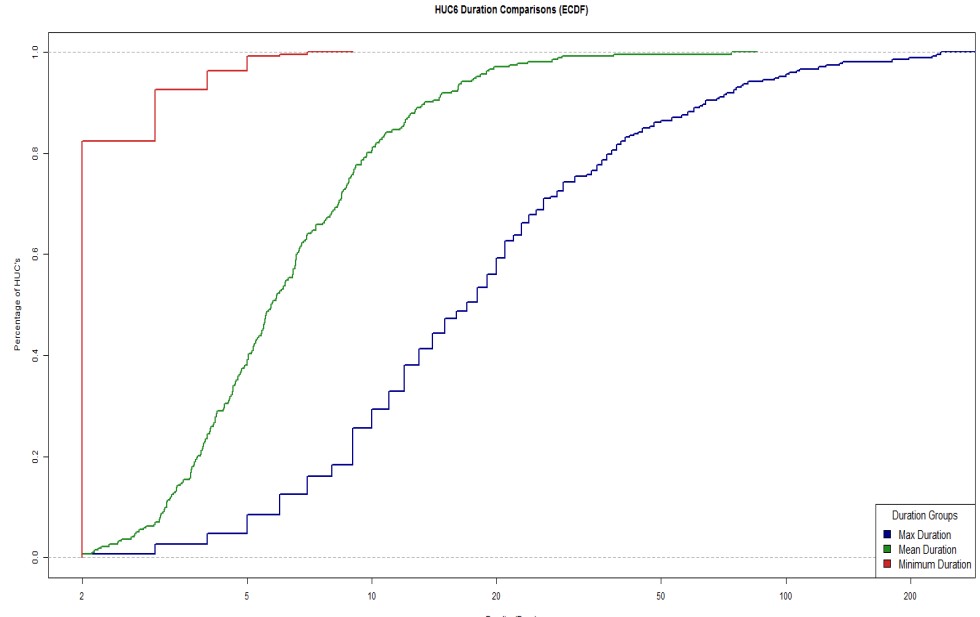

***Fig.13. HUC6 Event Duration CDF***

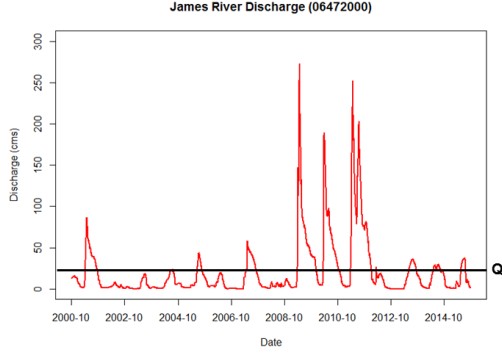
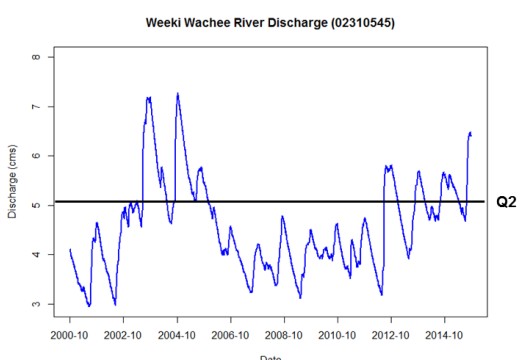

***Fig.14. Example Sites for Event Duration Concerns***





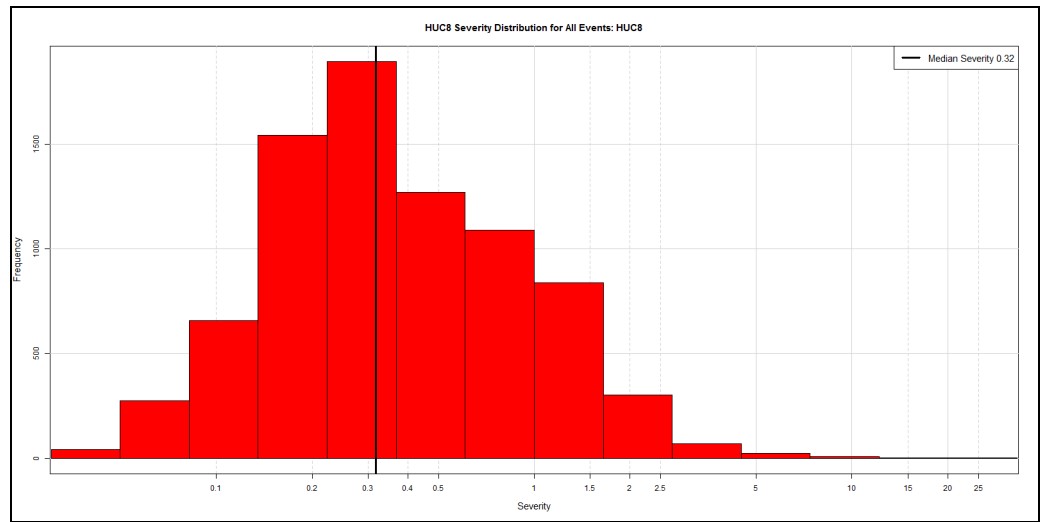

*Fig.15. Severity Score Distribution*

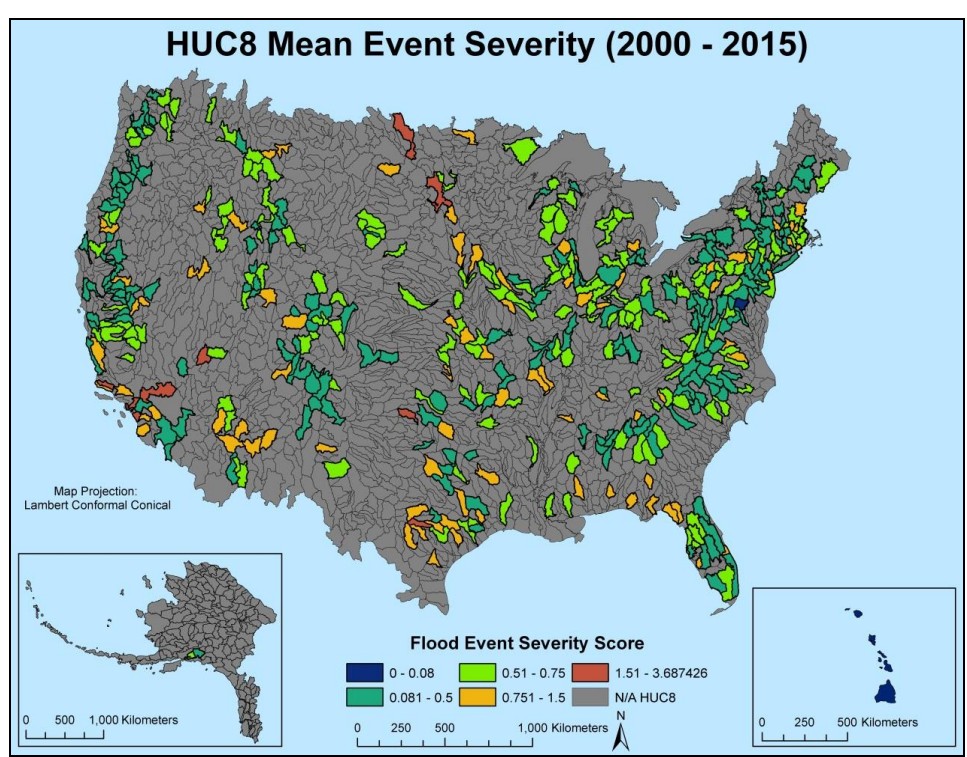

*Fig.16. Regional Distribution of Severity*



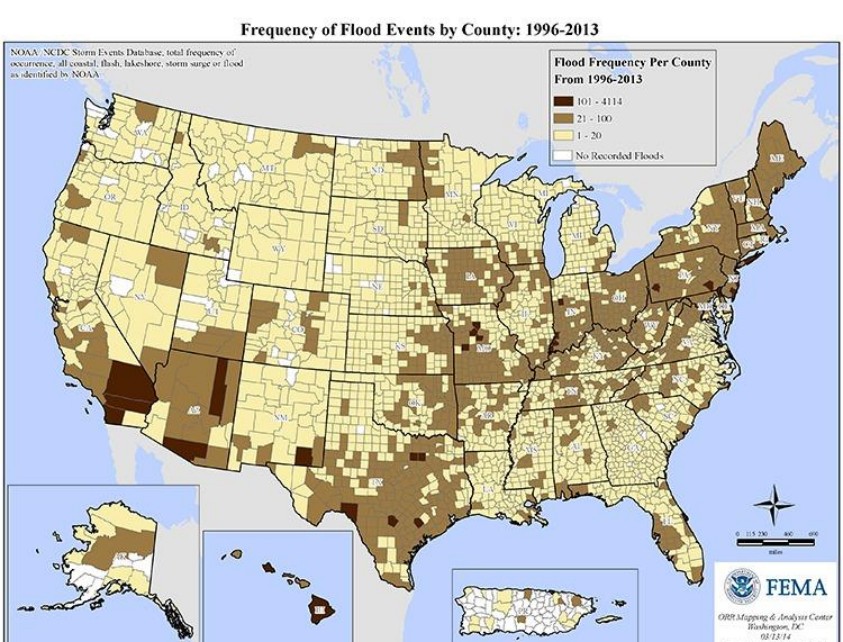

***Fig.17. FEMA Flood Frequency Estimates***

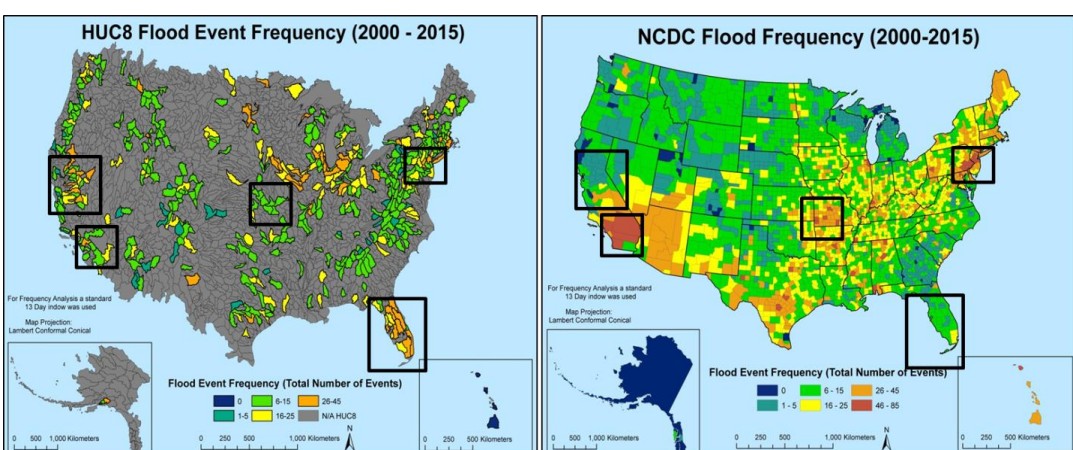

***Fig.18. Frequency Comparisons with a 13 Day Window (NCDC & Daily Discharge)***





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
