# Peer review of "Defining and Analyzing the Frequency and Severity of Flood Events to Improve Risk Management from a Reinsurance Standpoint"

_Hydrology and Earth System Sciences, 2017_

## Referee Comment (RC1) · Anonymous Referee #1 · 4 Sep 2017

This study applies a data driven approach to capture and aggregate multiple occurrences of flooding at various locations into independent events in attempts to solve the issues of inconsistent event definitions within the (re)insurance industry. The manuscript is well written and the results are valuable and interesting. Although the logic of this work is presently smoothly, the scientific novelty should be somehow emphasized in the manuscript. Some figures are repeated (see the comments below). Besides, I think the following major concerns of the work should be addressed before it gets published.

[Figure]

Lines 167 - 168, 8/6 digit HUC were chosen as they were felt to best represent how flood waves would impact a basin. So this is not a rigorous scientific conclusion and no other reference supports this?

Lines 201 - 204, and 208 - 209, if the method "was made to ensure that the threshold was not impacted by drastic variations within the annual maximum during a short period of time", then why "Sites with less than 5 years of data had their respective Q2 calculated from the annual maxima obtained through their daily discharge time series"? Isn't the methodology here contradictory to its purpose?

Lines 246 - 247, "the impacted area was defined as the number of sites within the desired HUC". This is questionable since the sites are not spatially uniformly distributed across all HUCs. But I see on lines 249 - 250, "severity was calculated by taking the sum of all scaled discharges and dividing by the total number of sites within the basin", which is fine.

Lines 266 - 267, authors said the frequencies within basins defined by the HUC6 are higher than frequencies defined by the HUC8. I think this is simply due to that the average area of each HUC6 is much larger than that of HUC8. Thus each HUC6 basin could contain several HUC8 basins (in the area) and generally has more flood events comparing to HUC8. But interestingly, authors found that "for each HUC, there was no interaction between the size of the catchment and the number of events" (lines 274 - 275). I hope the author could explain this obvious observation and the contradictory result.

Lines 280 - 281, "there does not appear to be a population bias throughout the study"? Please prove it, otherwise, do not make this conclusion.

Lines 285 - 292, Figures 12 and 13 are well explained by the numbers mentioned in this paragraph, but I think it would be more interesting to compare the shape of the corresponding CDF curves and it will be clear to see the difference of each duration within different HUCs.

Lines 303 - 304, "the duration of the events represent the observations at each site so based on our definition we can see long event durations". What does this sentence mean? What definition is it?

Line 367, can't find 071200 on Figure 5.

########### Figures ###########

Figure 4 left panel, Figure 9, Figure 16, and Figure 18 left panel are all the same. Figure 4 right panel is the same as Figure 11.

Can't recognize the numbers on Figure 5.

Figures 12 and 13 can be combined as one figure.

---

## Referee Comment (RC2) · Anonymous Referee #2 · 5 Oct 2017

General Comments:

This manuscript applied a data driven approach to solve the issues of inconsistent event definitions within the reinsurance industry. The methodology based on a peak over threshold (POT) was used to determine a metric for identifying independent peaks at various regions. Analyses were conducted on both HUC8 and HUC6 a total of 8,021 HUC8 events and 8,478 HUC6 events in the United States, in which each flooding event was characterized by duration, magnitude and severity. Although in my opinion the manuscript is well written and interesting to a broad readership, the following important

issues need to be addressed before acceptance of publication.

1. Threshold selection is the most important step of POT method, as it shows a large impact on estimation of the flood stage of that site, then affecting the estimates of flooding frequency and severity. The authors chose the median of annual maximums as the minimum threshold, which corresponds to the 2-year quantile. I am not sure if the 2-year quantile is suitable for all the sites of HUC8 and HUC6 basins, because it is only rough estimation for "bankfull discharge" on naturally occurring streams. Therefore, more information about how to select threshold should be given in this step. In addition, the introduction of calculation of 2-year quantile for sites with less or more than 5 years of data also needs more detail.

2. There is some discussion of threshold selection in Section 5 but it is very general and not quantitative. I suggest that the authors conduct a sensitivity statistical analysis on the threshold selected to test the impacts of changes in threshold on the frequency and severity across the basins. In addition, it is better to include and discuss your findings with the ones from the literature.

3. The number of figures must be reduced. Some similar figures may be combined as one figure for better comparison (e.g. for HUC8 and HUC6), such as Figures 2 and 3, Figures 4 and 5, Figures 6 and 7, Figures 8 and 10, Figures 9 and 11, and Figures 12 and 13.
* * *

---

## Author Comment (AC1) · 8 Nov 2017

Comment Line 167-168: Most papers that we used referenced in this paper specifically deal with European basins and they kept their basin areas smaller and applied size criteria. We wanted to get a better understanding Nationwide in the US and with basins varying drastically in characteristics and sizes we wanted to identify a common basin code to use. I did not come across any other paper that addressed Flood Frequency in the US using HUC's so we took it upon ourselves to choose two codes and work from there to identify which was more applicable. We chose HUC8 as our low end of our

size range because as you can see in Figure 1, we are able to get approximately 25% coverage of the US, based on the criteria we applied to select sites, had we moved to a HUC10 we would have only be able to cover ∼10% of the US which we felt was too low to accurately represent the US. With the HUC6 we were able to examine 85% coverage of the United States however we noticed the spacing of the sites might lead to more site events rather than events being attributed to the entire basin. With our goal being to identify events across a basin we wanted to eliminate the over estimation that you would see with more site events that should be aggregated to a single basin event.

Comment Line: 201-204 and 208 – 209: We recognize the error that we made in by contradicting ourselves in our methodology. We are in the process of removing those sites with < 5 years of data. The outcome of this removal does not affect the overall frequency that drastically as only a few sites within each basin level was removed to account for this error. We will recalculate all of the necessary statistics to account for this change as well.

Comment Line: 246 – 247 We clarified the language during this portion to be more representative of what we wanted to state. Impacted area was changed to affected sites. Affected sites, is more applicable because we wanted to avoid running into errors where it might be interpreted as the flooding extent in terms of area. The affected sites is still an indication of the severity as you noted in the comment because we scale the discharges at those sites and divide by the number of sites in the basin.

Comment Line: 266-267 While this is intuitive where you would think that a larger the basin the higher the frequencies would make sense. Due to the fact that we do not select every single site in the basins, we needed to plot the relationship between events per year and catchment area as well as events per year and site count. From those figures, there is not a significant relationship between catchment area (square km) and frequency in both the HUC6 and HUC8; however we do need to represent a better comparison to further prove our point that the HUC8 provides a better representation

of our method. The idea we painted with site count vs. frequency does show that there is a relationship between site count and frequency for both HUC6 and HUC8. The plot does show a stronger relationship for the HUC6; however this does not prove our hypothesis that the HUC8 is a better representation. We are currently working on a plot that examines the distribution of the percentage of sites impacted during the events that we record. We are going to show that with our method we are seeing fewer localized events that should be aggregated to basin events when our method is applied to the HUC8 when compared to the HUC6.

Comment Line: 280 – 281 This language has been removed from the paper. It was missed during our earlier revision as it is not referenced in any other capacity during this paper.

Comment Line: 285 – 292 The shape of the CDF will be discussed in the revisions to this paper.

Comment Line: 303 – 304 The language of this statement will be clarified in the revisions to this paper. The sentence was meant to describe that when we see these locations being effected by natural phenomenon's, our definition of event duration as described by the paper does not factor in these events with a drastic event generation caused by an ice jam or prolonged duration from intense ground saturation. We wanted to point out that while it may seem that the duration of these events is longer than what you would expect, they are accurate because of our Q2 threshold that there may be remnants of flooding that are not severe but are still being considered and aggregated to the same basin wide events.

Comment Line: 367 The figure was updated to increase the size of the labels as well as place a blue outline around the selected HUC6 and HUC8 so that it would be easier to identify which basins the figure was referring to.

Figure Comment: Figures were corrected to remove duplicates, increase label size and combined.

---

## Author Comment (AC2) · 8 Nov 2017

Comment 1: This comment was made by referee #1 and we have corrected it in the manor listed below. "We recognize the error that we made in by contradicting ourselves in our methodology. We are in the process of removing those sites with < 5 years of data. The outcome of this removal does not affect the overall frequency that drastically as only a few sites within each basin level was removed to account for this error. We will recalculate all of the necessary statistics to account for this change as well."

[Figure]

Comment 2: For this paper we wanted to give a basic understanding for our method of capturing flooding events within a basin on a nationwide approach. We selected the Q2 because of how previous literature states that it is the best approximation for "Bankfull Discharge." We recognize that this may be slightly low at certain sites as noted from your comment above. For the purpose of this paper we wanted to test how our methodology would work with a standard metric at first. We feel that the best way to capture the understanding of our method is to run our algorithm on a simple threshold first to get a sense of its applicability and then within a response paper and further dissection of this method we will run a sensitivity analysis and re-run the algorithm on the same subset of events with thresholds of Q5, Q10 and potentially higher thresholds. - This comment above is also a response to comment 1.

Comment 3: We made the necessary changes to the figures that we felt should be combined as noted. We also removed certain figures in response to the previous referee's comments to remove duplicate figures.

---

## Referee Report (RR1)

This paper analyzed the the Frequency and Severity of Flood Events of US. The topic is interesting and the results are helpful for risk management of the flood in US. However, there is some issues should be clearly addressed before it could be published in HESS. The comments are shown below:

Comments:

1.  The author indicates that the the maximum event duration for HUC8's was 232 days. There should be an explanation that why the duration was so long for that basin. And the frequency of flood seems not high. It seems that for some basins, threshold selection needs to be improved. Is there any possible methods for improvement? This seems not fully discussed in the paper.

2.  How to judge which basin size is best for the analysis of Frequency and Severity for the risk Management? It seems that there is not clearly explained in the method part.

3.  From duration analysis, HUC8 and HUC6, which is better?

4.  Line 9:HUC8 and HUC6 should be defined when it firstly appears.

5.  I suggest the author to separate section 5 into discussion part and conclusion part.

6.  Line 84-92: These sentences could be revised and the written should be more efficiency.

7.  Line 114-116: Add years of publication after "Mallakpour and Villarini" and "Black and Werrity".

8.  Line 163: The Pfafstetter Coding System is used and 6 levels are used for sub-basin identification. Why not other levels, for example, 5 levels or 7 levels.

---

## Author Response (AR2)

[revised manuscript text omitted]

**Comment [EM8]:** Added additional commentary on the selection of the Q2 and what steps will be taken to address the possible change in threshold based on Characteristics of the basin or a standard return period approach – Comment 1

characteristics. The final advantage to our method is that when looking at flood severity we do not look at exclusively magnitude but the addition of spatial extent adds an element to differences in severity regionally.

[revised manuscript text omitted]